# Language Modeling using LMUs and Implicit Self-Attention

## Abstract

Recent studies have demonstrated that the performance of transformers on the task of language modeling obeys a power-law relationship with model size over six orders of magnitude. While transformers exhibit impressive scaling, their performance hinges on processing large amounts of data, and their computational and memory requirements grow quadratically with sequence length. Motivated by these considerations, we introduce a novel attention module called implicit self-attention and construct a Legendre Memory Unit based model hat exhibits an $O(n)$ and $O(n \ln n)$ (or better) dependency for memory and computation respectively. Over three orders of magnitude, we show that for the same amount of training our model improves the loss over transformers about as much as transformers improve over LSTMs. Additionally, we demonstrate that adding global self-attention complements our architecture and the augmented model improves performance even further.

## 1 Introduction

Self-attention architectures such as the transformer (Vaswani et al., 2017) have been extremely successful in dealing with sequential data and have come to replace LSTMs and other RNN-based methods, especially in the domain of Natural Language Processing (Radford et al., 2018; 2019; Devlin et al., 2018). Transformers facilitate parallelization within training examples, and this allows them to fully leverage hardware accelerators such as GPUs, making training on datasets as large as 750GB feasible (Raffel et al., 2019; Gao et al., 2020). In addition to parallelizing training, self-attention architectures are much better at handling long-range dependencies relative to traditional RNNs, and this allows them to take advantage of context much longer than the $\sim$ 100-1000 tokens typical of RNNs, like the LSTM (Voelker & Eliasmith, 2018; Kaplan et al., 2020).

Transformers are general-purpose architectures that can be applied to a wide variety of problems and modalities. One of the drawbacks of such generality is the lack of *a priori* structure, which makes them heavily reliant on large quantities of data to achieve good results. Another limiting factor is that self-attention involves the computation of the attention matrix, $\boldsymbol{QK}^T$, which is of shape $n \times n$, with $n$ being the sequence length. Thus, transformer's compute and memory requirements grow quadratically with respect to the sequence length.

In this work, we explore a way of addressing these limitations. We base our approach on the non-parametric Linear Time-Invariant (LTI) component of the Legendre Memory Unit (Voelker et al., 2019). This LTI system, which we refer to it here as the LMU,[1] projects a sliding window of the input sequence onto Legendre polynomials to provide a temporal representation and compression of the input signal. Although the LTI system is an RNN, it has been shown to support both sequential and parallel processing of sequences (Chilkuri & Eliasmith, 2021). Another crucial component of our model is a modified attention mechanism that operates only on the output of the LMU at each time step, and not *across* time steps. The LMU state at each step captures information about the past tokens, and hence we call this attention mechanism *implicit self-attention*.

---

[1]We refer to the LTI system as the LMU as it is the distinguishing layer of our architectures. The original LMU was defined to include the LTI system as well as a subsequent nonlinear layer. We have essentially expanded this nonlinear layer to include a variety of familiar layers.

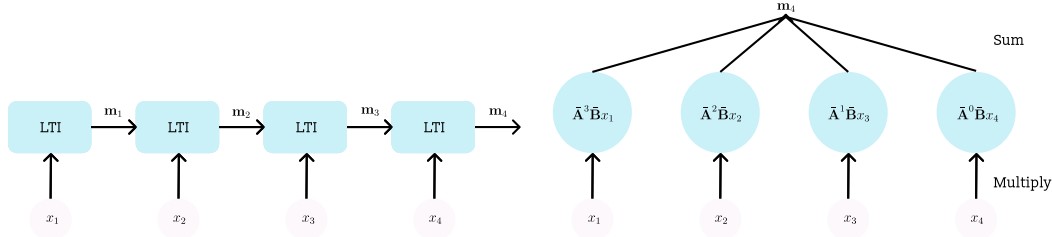

Figure 1: (left) Illustration of the standard sequential implementation for computing the hidden state $m_4$. The input $x_1$ is fed into the linear recurrent unit to compute the hidden state $m_1$, which, along with $x_2$, is then used to compute the next hidden state, and so on. (right) Illustration of the time-domain parallel implementation for computing the hidden state $m_4$. The inputs $x_1$-$x_4$ are used to compute the intermediate multiplies, which are then added together to compute the hidden state $m_4$, all without the need for any sequential operations.

Recent work (Kaplan et al., 2020; Gao et al., 2020) has explored the scaling properties of transformers in the context of autoregressive language modelling. They show that the performance of transformers scales as a power-law with model size (excluding embedding parameters), dataset size and the amount of compute used for training, when not bottlenecked by the other two. Inspired by these results, we validate our method by studying the scaling properties of our models. To that end, we start by reviewing some necessary background in Section 2, and then present the architectural details of our model in Section 4. In Section 5, we present our experiments with models containing up to 10 million parameters (or 1 million non-embedding parameters), which demonstrate a smooth power-law relationship with respect to the model size, and a loss that scales better than transformers. When the loss is matched to that of transformers, 10x fewer tokens are required.

## 2 BACKGROUND: THE LEGENDRE MEMORY UNIT

The non-parametric LTI component of the LMU (Voelker & Eliasmith, 2018) is the focus of our study. This LTI system is mathematically derived to project a sliding window of length $\theta$ of the input sequence onto $q$ Legendre polynomials. Thus, the two main hyper-parameters to choose when using it are $\theta$ and $q$. Naturally, if we desire to capture the fine-grained details of the input sequence, a large $\theta$, which sets the length of the sliding window, should be accompanied by a large $q$, the number of Legendre polynomials used in approximating the input. We present the state-update equations of the LTI component of the LMU below,[2]

$$m_t = \bar{A}m_{t-1} + \bar{B}x_t, \tag{1}$$

where the $\bar{A} = e^{A} \in \mathbb{R}^{q \times q}$ and $\bar{B} = A^{-1}(e^{A} - I)B \in \mathbb{R}^{q \times 1}$ matrices are frozen during training, with $A$ and $B$ defined as follows:

$$A_{i,j} = \frac{(2i+1)}{\theta} \begin{cases} -1 & i < j \\ (-1)^{i-j+1} & i \geq j \end{cases}, \tag{2}$$

$$B_i = \frac{(2i+1)(-1)^i}{\theta}. \tag{3}$$

Crucially, when needed, the LTI equation (1) above can be evaluated as a convolution, in parallel, as shown below (Chilkuri & Eliasmith, 2021):

$$m_t = \sum_{j=1}^{t} \bar{A}^{t-j} \bar{B}x_j, \tag{4}$$

or equivalently, defining

$$H = \begin{bmatrix} \bar{A}^0 \bar{B} & \bar{A}\bar{B} & \dots \end{bmatrix} \in \mathbb{R}^{q \times n}, \tag{5}$$

$$x = \begin{bmatrix} x_n & x_{n-1} & x_{n-2} & \dots & x_1 \end{bmatrix}^T \in \mathbb{R}^{n \times 1}, \tag{6}$$

---

[2]Focusing on one-dimensional inputs for now.

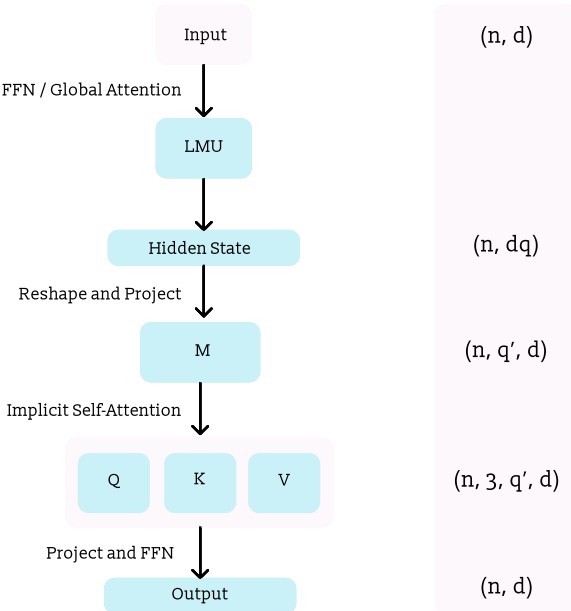

Figure 2: The LMU and implicit self-attention architecture along with output dimensions. In the illustration, $n$ refers to the sequence length, $q$ is the order and $q'$ is the reduced order, and $d$ is the embedding dimension. Normalization layers and skip connections are not shown. One variant uses the FFN component right after the input, and the other variant uses global attention.

the above convolution equation can be written as an element-wise multiplication in the Fourier space as follows:

$$\boldsymbol{m}_{1:n} = \mathcal{F}^{-1}\{\mathcal{F}\{\boldsymbol{H}\} \cdot \mathcal{F}\{\boldsymbol{x}\}\}. \tag{7}$$

## 3 RELATED WORK

Our work falls under the broad category of combining convolution with self-attention. Recent work has demonstrated that combining these two modules can be beneficial for language modeling, speech and other NLP applications (Yang et al., 2019; Wu et al., 2020; Gulati et al., 2020). For instance, Wu et al. (2020) introduce the *Long-short Range Attention* module that features a two-branch design, with the self-attention branch learning global interactions and the convolutional branch learning local interactions. Gulati et al. (2020) on the other hand propose a model that uses a single branch architecture, with the convolutional block connected directly to the attention block, and demonstrate improved performance on the task of speech recognition.

While our work is mathematically related to the models mentioned above, it differs from the previous approaches in three crucial ways: first, we do not learn the convolutional weights, but instead work with the analytically defined weights of the LMU; second, we introduce the novel implicit self-attention module that works on the hidden states of the LMU at each time-step; finally, we systematically study the scaling properties of our model in the context of language modeling.

Our model is also related to the many studies on reducing the quadratic computational and memory complexity of self-attention (Tay et al., 2020). Out of the several ways of improving efficiency, our work shares some similarity with the sliding window attention approach (Beltagy et al., 2020; Zaheer et al., 2020). While these methods use a form of masking of the full self-attention matrix to constraint attention to the $k$-nearest tokens, our technique relies on the LMU to compute optimal compressed representations of sliding windows of input vectors, each of which is then used as input to the implicit attention module.

## 4 ARCHITECTURE

In this paper, we modify the architecture presented in Chilkuri & Eliasmith (2021) to better deal with the task of language modelling, especially when the sequences are long and high-dimensional. Starting with the base LMU model, we describe the major components of our model below. An illustration of our architecture is presented in Figure 2.

**Memory Matrix**   Consider an input sequence $\{\boldsymbol{x}_1, \boldsymbol{x}_2, \ldots, \boldsymbol{x}_n\}$ of length $n$ where the individual elements are of dimension $\boldsymbol{x}_i \in \mathbb{R}^d$. The most natural way of using an LMU-based model on such sequences is to set $\theta = n$ and use an appropriately large $q$. The downside of using the LMU in such a manner, however, is that the hidden state of the LMU scales with input dimension and order: $\boldsymbol{m} \in \mathbb{R}^{dq}$. For example, in Section 5, we deal with $n = 1024$ and $d$ that is as large as 204. Even when using a small $q$ of 100, we may end up with hidden states that are as large as $\mathbb{R}^{20k}$, which is highly undesirable.

One way around this issue is to take inspiration from standard convolutional network architectures (CNNs) and work with a smaller sliding window, $\theta \approx 10$, which in turn allows us to use a small LMU order, $q \approx 5$, thus taming the hidden state dimension (see Chilkuri & Eliasmith (2021) for more details). However, enforcing a small sliding window prompts the use of many stacked LMU layers in order to increase the 'receptive field' (or the effective $\theta$) of the model, very similar to how CNNs often use small kernels with many convolutional layers. Unsurprisingly, such an approach results in very deep models, which can be problematic to train.

Here, we choose to follow the middle path, i.e, $0 \ll q \ll n$, but instead of working directly with the potentially high-dimensional hidden state $\boldsymbol{m}$, we disentangle the input dimensions from the order. In other words, we perform our operations on the matrix $\boldsymbol{M} \in \mathbb{R}^{d \times q}$ and not on the vector $\boldsymbol{m} \in \mathbb{R}^{dq}$. This is beneficial because while a fully-connected layer needs $d^2 \cdot q^2$ parameters to process the $\boldsymbol{m}$ vector, processing the matrix $\boldsymbol{M}$ with the help of two fully connected layers, one for the rows and one for the columns, requires only $d^2 + q^2$ parameters.

**Implicit Self-Attention**   The main feature distinguishing our architecture from past work is the LMU. As mentioned above, the output of the LMU layer compresses past history at each time-step, which is captured by the $\boldsymbol{M} \in \mathbb{R}^{q \times d}$ matrix. Our modified self-attention acts on this matrix to combine temporal information. As a result, self-attention does not act directly on the input sequence, but rather on a compressed version of the input, which is available at each moment in time and covers a window, determined by $\theta$. Ignoring the bias vectors, normalization layers, and skip-connections, we first execute the following sets of operations simultaneously:

$$\boldsymbol{Q} = \sigma(\boldsymbol{L}_1 \boldsymbol{M}) \qquad\qquad \boldsymbol{K} = \sigma(\boldsymbol{L}_2 \boldsymbol{M}) \qquad\qquad \boldsymbol{V} = \sigma(\boldsymbol{L}_3 \boldsymbol{M}), \qquad (8)$$

where $\boldsymbol{L}_i \in \mathbb{R}^{q' \times q}$, $\sigma$ is a non-linearity such as `gelu`, and the matrices $\boldsymbol{Q}, \boldsymbol{K}, \boldsymbol{V}$ are all in $\mathbb{R}^{q' \times q}$. The primary motivation behind this transformation was to reduce the size of the matrix dimension, and we have found the setting $q' = q/10$ to work well in our experiments, thus the attention matrices contain far fewer elements than $n^2$. For example, in our largest model we set $q = 250$, resulting in $q' = 25$.

Following the computation of $\boldsymbol{Q}, \boldsymbol{K}$, and $\boldsymbol{V}$ two additional computations result in a $d$-dimensional vector $\boldsymbol{m}$:

$$\boldsymbol{M}' = \mathrm{softmax}(\boldsymbol{Q}\boldsymbol{K}^T)\boldsymbol{V}, \qquad (9)$$

$$\boldsymbol{m} = \boldsymbol{p}\boldsymbol{M}', \qquad (10)$$

where $\boldsymbol{p} \in \mathbb{R}^{1 \times q'}$.

In practice, equation (8) can be made far more efficient computationally, especially during inference, by following the recipe outlined in Section A.1.

**Feedforward Network**   We have also found it beneficial to include a feedforward network (FFN) (Vaswani et al., 2017) before the LMU and after the implicit self-attention block. The FFN component is defined below:

$$\boldsymbol{y}(\boldsymbol{x}) = \sigma(\boldsymbol{x}\boldsymbol{W}_1 + \boldsymbol{b}_1)\boldsymbol{W}_2 + \boldsymbol{b}_2, \qquad (11)$$

where $\boldsymbol{W}_1 \in \mathbb{R}^{d \times d'}$, $\boldsymbol{W}_2 \in \mathbb{R}^{d' \times d}$, $\boldsymbol{b}_1 \in \mathbb{R}^{d'}$ and $\boldsymbol{b}_2 \in \mathbb{R}^d$.

Table 1: Memory and computation scaling with sequence length during training and inference.

| Layer | Memory | Compute |
|---|---|---|
| Full Attention | $O(n^2)$ | $O(n^2)$ |
| LMU (Parallel) | $O(n)$ | $O(n \ln n)$ |
| LMU (Recurrent) | $O(1)$ | $O(n)$ |

Table 2: Parameter counts and compute (forward pass) for one layer of the network, per token. The first row indicates the number of FLOPs when following the implementation in Section A.1. Additional background information regarding various implementations of the LMU is provided in Appendix A.2.

| Operation | Parameters | FLOPs per Token |
|---|---|---|
| $LMU + Q + K + V$ | $3qq'$ | $3d\left[5(q'+1)(\log_2 n + 1) + 6q'\right] + 6qq'$ |
| $QK^T$ | – | $2dq'^2$ |
| $M'$ | – | $2dq'^2 + dq'$ |
| $m$ | $q'$ | $2dq'$ |
| FFN | $2dd'$ | $4dd'$ |

**Global Self-Attention** We also explore the use of a global self-attention block in place of the FFN component before the LMU. We find that introducing this layer, while computationally expensive, further improves the cross-entropy score. We believe that the improvement comes from the fact the LMU and self-attention are complementary: the LMU's implicit self-attention is good at prediction with limited context, and the traditional self-attention captures long-range dependencies. We wish to explore the use of efficient self-attention blocks – which scale better than $O(n^2)$ – in the future.

**Complexity** As shown in Table 1, our architecture employing the parallel LMU along with implicit self-attention has memory requirements that are linear with respect to the sequence length, and it has computational requirements that also grow as $n \ln n$. When we use the recurrent version of the LMU, the memory and compute requirements scale as $O(1)$ and $O(n)$ respectively. Recurrent implementations, while not as efficient on GPU architectures for large batch sizes, are ideally suited to edge applications, especially with efficient hardware support. Notably, if we add global attention to our model, then both compute and memory become quadratic, just like the original transformer.

We also list the number of floating point operations per-token for the (parallel) LMU model in Table 2.

## 5 EXPERIMENTS

**Dataset** We train our models on the publicly available internet text dataset called OpenWebText2 (OWT2).[3] Similar to the WebText2 dataset (Radford et al., 2019), OWT2 was created using URLs extracted from Reddit submissions with a minimum score of 3 as a proxy for quality, and it consists of Reddit submissions from 2005 up until April 2020. After additional filtering, applying the pre-trained GPT2 tokenizer containing 50257 tokens (Radford et al., 2019) results in approximately 8 billion tokens in total. We use a train/validation/test split of 96/3/1%.

**Training Details** We train our models in an autoregressive manner using the Adam optimizer with all the default settings. We use sequences containing 1024 tokens, and in cases where the documents have fewer than 1024 tokens, we pack multiple documents into the same sequence, separated by the `<|endofsequence|>` token. We use a learning rate schedule with a linear warmup and cosine decay to zero, while also reducing the learning rate on plateau. We chose to train our models to

---
[3]https://www.eleuther.ai/projects/open-web-text2/

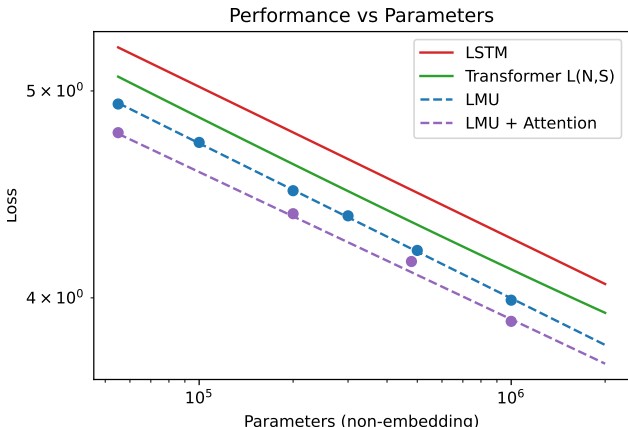

Figure 3: Cross-entropy scores in nats, averaged across all the tokens in the sequence. Transformers and LSTMs fits are from Kaplan et al. (2020). Our models perform better than Transformers and LSTM models up to 1 million non-embedding parameters.

process a maximum of 13 billion tokens; at a batch size of 512, this amounts to training for 25000 steps. Additionally, one of the important considerations when doing NLP experiments is the size of the embedding vectors, $d$. In this work, in order to facilitate a fair comparison to the transformer models (Kaplan et al., 2020), we use the following rule to determine $d$:

$$d = \sqrt{\frac{N}{24}},$$

where $N$ represents the number of non-embedding (and trainable) parameters. We also note that we attempted to match compute across all experiments to the corresponding transformer models.

**Results**    Here we present the results of our experiments that use the LMU architecture described above with the non-embedding (and trainable) parameters ranging from 55k to 1M (i.e., from 2.5M to 10M, if we include all parameters). The cross-entropy results are presented in Figure 3. For the transformer models, we list the scores obtained by using the following power-law fit,

$$\text{Transformer}(N, S) = \left(\frac{N}{6.5 \cdot 10^{13}}\right)^{-0.077} + \left(\frac{S}{S_{min}(S)}\right)^{-0.76}, \tag{12}$$

where $N$ refers to the number of non-embedding parameters and $S$ is the total number of training steps (set to 25000 at batch size of 512, or equivalent). The power-law is obtained from Kaplan et al. (2020); the fits were generated by training several transformer models, ranging in size from 768 to 1.5 billion non-embedding parameters, on OpenAI's WebText2 dataset.[4] In addition, we compare against the power-law for LSTM models, also from Kaplan et al. (2020), that use 10x more training steps than the transformer and LMU models:

$$\text{LSTM}(N) = \left(\frac{N}{7.45 \cdot 10^{14}}\right)^{-0.071}.$$

Similar to the transformer and LSTM models, we notice that the performance of our models depends strongly on scale, with the loss of the LMU model exhibiting the following power-law relationship with respect to $N$:

$$\text{LMU}(N) = \left(\frac{N}{1.95 \cdot 10^{14}}\right)^{-0.072}.$$

---

[4]We note that the LMU models do not make use of positional embedding parameters, and for the transformer models, the parameter count excludes the positional embedding parameters (given by embed dimension * sequence length). For our largest model, the total number of non-parametric weights inside all the LMU layers (given by order * order + order) amounts to about 70% of the positional embedding parameters.

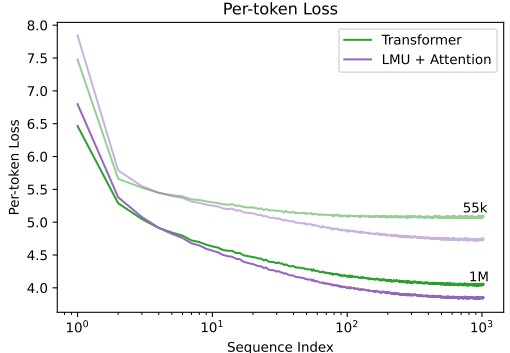 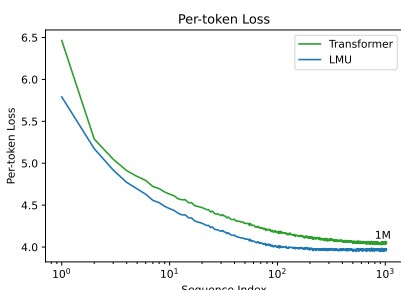

Figure 4: (left) Comparison of per-token loss of an LMU model (with global attention) and a transformer model. (right) Per-token loss of an LMU model (without global attention) alongside the transformer's loss.

The LMU model with global self-attention scales as follows:

$$\text{LMU}_G(N) = \left( \frac{N}{3.80 \cdot 10^{14}} \right)^{-0.069}.$$

It remains to be seen whether our models retain this performance advantage when $N \gg 10^6$.

The utility of adding global attention becomes clear when we observe the per-token loss plots of the three models. In Figure 4 (right), we notice that although the LMU's per-token loss is better overall than the transformer's, it flattens relatively early, around 100 tokens, suggesting that implicit attention alone does not capture long context. The LMU and Attention model, on the other hand, continues improving with increasing context, similar to the transformer.

It is also interesting to note that we can compare LMUs and transformers by determining approximately how much training is required for a transformer to match LMU loss. Figure 5 demonstrates that our models, trained on 13 billion tokens, have similar scaling to transformers trained on 130 billion tokens. Consequently, the LMU architecture is 10x more data efficient. In addition, our LMU models with global attention continue to outperform transformer models trained on 10x more tokens (or with 10x more training steps) by a significant margin.

## 6 DISCUSSION

Semi-supervised learning has proven to be a very effective technique in Natural Language Processing. General purpose language models pre-trained on a large corpus of text in an unsupervised manner and fine-tuned on tasks such as sentiment analysis and question answering often outperform highly task-specific architectures that receive no pre-training. The performance of models on the task of language modelling is thus a crucial metric that is indicative of the downstream performance of such models on a slew of tasks involving natural language.

While the performance of our models on the task of language modelling suggests an interesting trend, due to the scale of our experiments however, we do not consider this to be definitive evidence for the superiority of our LMU architecture. As a result, a core objective for future research is to show that the observed trends hold over 6 orders of magnitude, as demonstrated by Kaplan et al. (2020) for transformers.

Additionally, we would like to point out three things in relation to the use of OpenWebText2: 1) We made sure to construct or filter the OpenWebText 2 dataset to resemble Open AI's WebText 2 dataset as closely as possible. 2) We trained transformer models on our version of the dataset and confirmed that the final performance is lower bounded – lower the better – by the power-law and scores reported in Kaplan et al. (2020). 3) Our experiments are reinforced by the results listed in Gao et al. (2020) and Kim (2021), where the authors use OpenWebText2 to study the scaling properties of GPT-2-like transformers.

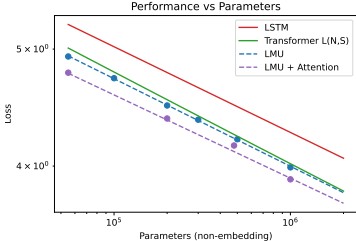

Figure 5: Approximately matching the loss between transformers and LMUs requires 10x more training for the transformer. The LMU and Attention model continues to significantly outperform transformers with 10x less training.

Regarding the use of the original variant of the transformers as the point of comparison, we note that while efficient transformers (Zaheer et al., 2020; Beltagy et al., 2020; Wang et al., 2020), thanks to their lower memory requirements, can be trained on batches of extremely long sequences and this proves advantageous for tasks that benefit from long context, such as summarization and question answering. While the efficient transformers tend to outperform the original transformer in such settings, the original variant maintains better performance when working with typical sequence lengths. Since we are dealing with sequences of length 1024, we reasoned that it would be most appropriate to compare against models employing full self-attention and not an approximate version. In short, we are demonstrating advantages where efficient transformers do not.

## 7 CONCLUSION

In this work, we employ the Legendre Memory to construct a model that is well-suited to handling long sequences with high-dimensional elements. We apply our architectures to model natural language in the infinite data limit, demonstrating that: (1) like the established architectures such as transformers and LSTMS, our models also exhibit a power-law relationship between the cross-entropy loss and model size; and (2) at the small-medium scale, our models have better scaling properties than other approaches.

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

## A  APPENDIX

### A.1  REDUCED-ORDER LMU

When implementing the LMU in these models, we use a parallelizable approach that computes the impulse responses of the LMU (which are essentially the Legendre polynomials), and convolve those with the input sequences (either using raw convolution or FFT-based convolution). Specifically, given the $q$-dimensional impulse response $\boldsymbol{H}$, we compute the LMU memory state $\boldsymbol{M} \in \mathbb{R}^{d \times q}$ at the current time as

$$\boldsymbol{M} = \boldsymbol{X} * \boldsymbol{H} \tag{13}$$

where $\boldsymbol{X} \in \mathbb{R}^{n \times d}$ is the time-series of previous inputs to the LMU, $\boldsymbol{H} \in \mathbb{R}^{q \times n}$ is the LMU impulse response, and $*$ is the convolution operator.

We have found that our models are most expressive when using a value of $q$ that is significantly larger than $q'$, as this allows the LMU to "remember" the time history with high fidelity, but only use the parts of the history that are most relevant. Rather than explicitly computing the full LMU output $\boldsymbol{M}$ and then reducing this with the $\boldsymbol{L}_i$ transformations as per equation (8), we propose applying the $\boldsymbol{L}_i$ transformations directly to the impulse responses

$$\tilde{\boldsymbol{H}}_i = \boldsymbol{L}_i \boldsymbol{H} \tag{14}$$

and then applying these individually to directly compute $\boldsymbol{Q}$, $\boldsymbol{K}$, and $\boldsymbol{V}$:

$$\boldsymbol{Q} = \sigma(\boldsymbol{X} * \tilde{\boldsymbol{H}}_1) \qquad \boldsymbol{K} = \sigma(\boldsymbol{X} * \tilde{\boldsymbol{H}}_2) \qquad \boldsymbol{V} = \sigma(\boldsymbol{X} * \tilde{\boldsymbol{H}}_3) \tag{15}$$

This is mathematically equivalent to the formulation expressed previously, but uses significantly fewer operations per token, particularly when $q$ is large or the ratio of $q'$ to $q$ is small:

$$\mathfrak{L}_{\text{LMU+Q+K+V}} = 3[d\mathfrak{L}_{\text{FFT}}(q') + 2qq'] \tag{16}$$

where $\mathfrak{L}_{\text{FFT}}(q')$ is given by Equation 19 with $q \to q'$.

### A.2  LMU IMPLEMENTATION TRADE-OFFS

The LMU itself is a LTI dynamical system, with a number of options for implementation. One implementation is to perform the update each timestep in state-space, using state-space matrices discretized using the zero-order hold (ZOH) method for high accuracy. The operations required (per LMU layer and per token) are the multiplications by the $\bar{\boldsymbol{A}}$ and $\bar{\boldsymbol{B}}$ matrices (with number of elements $q^2$ and $q$, respectively):

$$\mathfrak{L}_{\text{SS}} = 2d(q^2 + q). \tag{17}$$

Another option is to use an explicit Runge-Kutta method to update the LMU states. By taking advantage of the unique structure of the $\boldsymbol{A}$ and $\boldsymbol{B}$ matrices (Equations 2 and 3), this implementation is able to reduce the complexity from $O(q^2)$ to $O(q)$, requiring the following approximate number of operations:

$$\mathfrak{L}_{\text{RK}} = 6rdq. \tag{18}$$

where $r$ is the order of the Runge-Kutta method. The disadvantage to this option is that it does not implement the exact same dynamics as the ideal system discretized with ZOH, and is less numerically stable particularly for higher values of $q$.

A disadvantage to both these options is that they must update LMU states sequentially, which is particularly ill-suited when using highly parallel hardware (e.g. GPU) with a long sequence of inputs available. In this case, we can take the impulse response of the LMU system (discretized with

ZOH), and convolve it with an input in the FFT domain. This implements the exact same dynamics as the ZOH state-space system, but with a complexity that is $O(q)$ rather than $O(q^2)$:

$$
\begin{aligned}
\mathfrak{L}_{\text{FFT}} &= \frac{d}{n} \left[ C(2n) + c_m qn + qC(2n) \right] \\
&= d \left[ 5(\log_2 n + 1)(q + 1) + 6q \right].
\end{aligned} \tag{19}
$$

Here, $C(n)$ is the number of FLOPs for a radix-2 Cooley-Tukey FFT implementation (Johnson & Frigo, 2012):

$$
\begin{aligned}
C(n) &= 2C\left(\frac{n}{2}\right) + \frac{n}{2}(c_m + 2c_a) \tag{20} \\
&= 5n \log_2 n \tag{21}
\end{aligned}
$$

where $c_m = 6$ is the number of FLOPs per complex multiply, and $c_a$ is the number of FLOPs per complex addition. For our standard sequence length of $n = 1024$, this results in:

$$
\mathfrak{L}_{\text{FFT}-1024} = d(61q + 55). \tag{22}
$$

## A.3 LMU MODEL DETAILS

Table 3: LMU model details. $N$ refers to the number of non-embedding and trainable parameters; $d$ is the embedding dimension; $q$ and $\theta$ define LMU's order and the length of the sliding window; $q'$ is the number of rows in the $\boldsymbol{L}$ matrix. We adjust the post-FFN inner ration to obtain the right parameter count for comparison to transformers – it's usually set to something in between 1.9 and 2.1.

| $N$ | Total Parameters | $d$ | $q$ | $q'$ | Layers | $\theta$ | Pre-FFN Inner Ratio |
|------|------|------|------|------|------|------|------|
| 55k | 2.4M | 48 | 50 | 5 | 3 | 350 | 1.5 |
| 100k | 3.3M | 65 | 65 | 7 | 3 | 350 | 1.5 |
| 200k | 4.8M | 91 | 90 | 9 | 3 | 350 | 1.5 |
| 300k | 5.9M | 112 | 110 | 13 | 3 | 350 | 1.5 |
| 500k | 7.3M | 144 | 150 | 15 | 3 | 350 | 1.5 |
| 1M | 11M | 204 | 220 | 22 | 3 | 350 | 1.5 |

