# OpenReview forum: "Language Modeling using LMUs: 10x Better Data Efficiency or Improved Scaling Compared to Transformers"
_ICLR.cc/2022/Conference — ICLR 2022 Submitted_

### Official Review · Reviewer_9uKE · 2021-10-26

**Correctness:** 2
**Technical Novelty And Significance:** 3
**Empirical Novelty And Significance:** 3
**Recommendation:** 5
**Confidence:** 3

**Main Review:**

The quadratic memory and time complexity of transformers have resulted in recent efforts to replace the softmax self-attention with efficient variants. This paper presents one such effort, which is very interesting, and could pave the way for a new generation of transformers. The approach is simple and relatively straightforward, and the writing is overall clear.

I have two main concerns about the paper that prevent me from recommending acceptance, both boil down to the same point: I am not convinced that the experiments support the authors interpretation. First, the paper focuses on the main idea of using LMU, but the proposed model contains many different components, that are orthogonal to it. For instance, why is the $L_i$ projection and the following non-linearity needed? Would applying it to the transformer lead to similar gains? The authors say _Our modified self-attention acts on this matrix to combine temporal information. As a result, self-attention does not act directly on the input sequence, but rather on a compressed version of the input._ But the same compression can be applied on standard SA as well. Further, the FFN layer after the input was found to be useful by the authors. Did they also add it to their baselines? In general, some ablations would make the paper much stronger and shed more light on the proposed model (on top of what is mentioned above, I would also like to see the effect of the values of q and q').

Second, the experimental setup is underspecified: the paper's main selling point is the strong performance compared to the transformer, but important details are missing with respect to both the models and the baselines. How many layers does each model have? What is the hidden size? Importantly, the last experiment (Fig. 5), which appears in the paper title, is even less specified: was the same transformer model trained on 130GB of data? Where did this data come from? As a side note, it is also unclear that such small model could leverage 130GB of data, and thus this experiment might not be ideal to test the sample efficiency of the proposed model. I would recommend providing more details about the experimental setup, and justifying the experimental choices.


**Summary Of The Paper:**

This paper proposes an efficient approximation of the transformer model. The authors replace the matrices in the dot product with a the output of a Legendre Memory Unit (LMU) transformation of the input, which projects a sliding window of the input sequence onto Legendre polynomials. This projection is fixed and not learned, which allows for memory savings. The resulting matrices are also smaller, which leads to further efficiency gains. The idea of using LMUs has been proposed before, and this paper applies them to the transformer. Experiments with language modeling show substantially lower (i.e., better) loss compared to the vanilla transformer model on several model sizes.


**Summary Of The Review:**

The paper presents a very interesting idea with strong empirical results on language modeling, that could inspire a new generation of transformers. However, there is some uncertainty with respect to the connection between the experiments and the research hypotheses, which makes it challenging to convince oneself of the value of the results. I am looking forward to reading the authors' response.

---

> ### Author Response · Authors · 2021-11-22
> **Response 4**
>
> We thank the reviewer for their comments.
>
> >> First, the paper focuses on the main idea of using LMU, but the proposed model contains many different components, that are orthogonal to it. For instance, why is the  projection and the following non-linearity needed?
>
> We agree with your comment; our model is incomplete without the implicit attention module. Both LMU and implicit self-attention are central to the models we have built, and the content of the paper now better reflects that.
>
> Regarding L_i projection, the motivation behind that was to simply reduce the size of the matrix dimension from q to q’, with q’ = q/10, so that the attention matrices have 100 times fewer elements. And since we were already performing an affine transformation, we thought that taking into account multiplicative interactions using a non-linearity wouldn't hurt.  We now explain this in more detail in the paper.
>
> >> Second, the experimental setup is underspecified: the paper's main selling point is the strong performance compared to the transformer, but important details are missing with respect to both the models and the baselines. How many layers does each model have? What is the hidden size? Importantly, the last experiment (Fig. 5), which appears in the paper title, is even less specified: was the same transformer model trained on 130GB of data? Where did this data come from?
>
> We have added in additional model details. Please see Table (3) in the paper.
>
> The plot for the transformer in Figure 5 was generated using the power-law described in the paper by Kaplan et al. The fit was generated by training several transformer models, ranging in size from 768 to 1.5 billion non-embedding parameters, on OpenAI’s WebText2 dataset.
>
> In the same paper, they also describe a more general power-law that is a function of the model size and the number of tokens processed; this more general fit was used in creating the plot in Figure 3 of our work (by setting tokens processed = 13 billion).  We have updated the paper to include this additional information.

---

### Official Review · Reviewer_ooZ6 · 2021-11-02

**Correctness:** 2
**Technical Novelty And Significance:** 2
**Empirical Novelty And Significance:** 1
**Recommendation:** 3
**Confidence:** 4

**Details Of Ethics Concerns:**

There is no ethical statement in the paper. Given that it deals with language models it would be useful to discuss what are the implications of training on web text and other potential biases that can be encoded in pretrained language models.

**Main Review:**

Strengths
* Improving the data efficiency in language models is an important problem that so far studies have shown that can be achieved by scaling the size of the model. Having additional ways to improve data efficiency by changing the model design is definitely of interest.
* Using the Legendre Memory Unit to substitute self-attention in transformers is interesting and has several potential merits: it can reduce the complexity and does not increase the size of the layer.  Although it turns out that additional components need to be introduced for good performance.


Weaknesses
* Something that stands out is the lack of discussion and comparison to related works that employ recurrent formulations of attention. Kernel-based variants of self-attention have a recurrent formulation and lead to linear complexity (see [1,2,3,4]).
* The rationale behind the architectural choices for the self-attention component is not well explained or empirically verified. I understand that LMU might have limited capacity but this is not specifically discussed and it is unclear how each component contributes to the end performance.
* Even though this paper proposes a new efficient transformer, the evaluation does not focus on computational efficiency aspects and comes across as incomplete. Quantifying the latency and scaling with respect to sequence length would be needed for convincing the reader about its usefulness.
*  The evaluation focuses on comparing with an empirical law learned on a different experimental configuration and there is a concern about how comparable are the results to the ones obtained in this study and the validity of the conclusions. I don't think it is good practice to use these power laws universally, especially for this purpose. Another limitation is that it is unclear whether the improvement would hold when the size of the model increases; the evaluation is dealing with scaling laws after all. I would suggest the following:
  * To make the comparison more fair, I would suggest to train transformer models with varying size and derive a power law based on the exact same experimental configuration used for LMU models.
  *  The claims regarding the 10x better data efficiency are not well supported and I would suggest the authors to compare with transformer models on standard LM benchmarks and potentially to some downstream tasks such as MT to make a stronger case.



Questions:
* How can we be sure that the specific power law derived from a different experimental configuration in Kaplan et al. (2020) corresponds to the empirical law that can be derived for transformers in this particular evaluation setting? It would be great if the authors discuss this or provide some supporting evidence about its correctness.
* From the evaluation, it seems that comparing to other efficient transformers was not the main focus of the work but people might wonder how well it works compared to alternatives. What recommendation the authors would give for those interested in using it?
*  What is the contribution of each specific design choice, such as the FFN/global attention and implicit self-attention, to the end performance? An ablation analysis would be most appropriate for quantifying this.
* Could the authors provide more details about the evaluation settings and number of parameters of each model for the per-token loss comparison? Is the per-token loss computed on the test set or training set?  Why is the sequence index capped at 10^3?

[1] https://arxiv.org/pdf/2006.16236.pdf

[2] https://arxiv.org/pdf/2009.14794.pdf

[3] https://openreview.net/pdf?id=QtTKTdVrFBB

[4] https://openreview.net/pdf?id=ot2ORiBqTa1

**Summary Of The Paper:**

This paper aims to address two issues with standard transformers for language modeling, namely their large data requirements and their computational cost due to self-attention complexity. The main proposal is to replace self-attention in transformers with Legendre Memory Units from Voelker et al (2019) which reduce memory complexity to linear and computation to log linear for their convolutional variant as well as  constant/linear for their recurrent variant respectively. The empirical claim that it makes is that the proposed transformers require 10x fewer tokens to reach the same loss, when comparing to the laws derived from the experimental configuration of Kaplan et al (2020).

**Summary Of The Review:**

Overall, the idea is interesting and the goal of the paper is definitely valuable. The framing and the delivery, however, are greatly lacking. My main concerns are regarding the validity of the evaluation and the lack of experimental details and comparison to alternative efficient variants. For these reasons, I think that the paper is not ready yet because it lacks sufficient evidence for the main claims and requires additional experimental efforts to better demonstrate the effectiveness of the proposed method.

---

> ### Author Response · Authors · 2021-11-22
> **Response 3**
>
> We thank the reviewer for their comments.
>
> >> How can we be sure that the specific power law derived from a different experimental configuration in Kaplan et al. (2020) corresponds to the empirical law that can be derived for transformers in this particular evaluation setting? It would be great if the authors discuss this or provide some supporting evidence about its correctness.
>
> We apologize for any confusion here, and we would like to point out three things:
> 1) We made sure to construct / filter the OpenWebText 2 dataset to resemble Open AI’s WebText 2 dataset as closely as possible.
> 2) We trained transformer models on our version of the dataset and confirmed that the final performance is lower bounded -- lower the better -- by the power-law and scores reported in Kaplan et al.
> 3) Our experiments are reinforced by the results listed in [2, 3], where the authors use OpenWebText2 to study the scaling properties of GPT-2-like transformers.
>
> [1] Scaling Laws for Neural Language Models, Kaplan et al, 2020
> [2] The Pile: An 800GB Dataset of Diverse Text for Language Modeling, Gao et al, 2020
> [3] Scaling Laws for Language Transfer Learning, Christina Kim, 2021
>
> In short, we went to significant lengths to reproduce the experimental setting of Kaplan et al, and verify that we can train transformers to the same level of performance before introducing the LMU architecture. We have clarified this in the new version of the paper.
>
> >> Could the authors provide more details about the evaluation settings and number of parameters of each model for the per-token loss comparison? Is the per-token loss computed on the test set or training set? Why is the sequence index capped at 10^3?
>
> The plots in Figure 4 were created using models containing 55k and 1M non-embedding parameters. The per-token loss computation follows the same recipe as the cross-entropy loss, except that we do not compute the mean across all time-steps. We used the test set to obtain these plots. Our models were trained on sequences containing 1024 tokens, and so we capped the index there.

---

> > ### Comment · Reviewer_ooZ6 · 2021-11-30
> > **Response**
> >
> > Thanks for the reply and clarifications! Discussing the empirical rationale more would help but I think that, overall, the claims made in this paper are not well supported and leave open questions.
> >
> > Also, demonstrating improvements where existing efficient transformers do not is not a reason to avoid comparisons with them, especially when introducing a new variant.  A more focused framing on scaling behavior would help perhaps make these arguments more convincing.
> >
> > For these reasons, I am inclined to maintain my initial recommendation.

---

### Official Review · Reviewer_HHvw · 2021-11-03

**Correctness:** 2
**Technical Novelty And Significance:** 3
**Empirical Novelty And Significance:** 2
**Recommendation:** 3
**Confidence:** 4

**Main Review:**

Strengths:
1. Novel and interesting idea, well motivated architecture.
2. Empirical evaluation over a wide range of model sizes in terms of # parameters.

Major limitations:
1. The paper presents an incomplete comparison that plays to the strength of the LMU architecture by comparing only parameter-matched models. Given that a significant fraction of the LMU compute is hidden in non-trainable parameters, a fair comparison should also compare models trained with an equivalent amount of training computation; ideally in terms of total training compute used.
2. Limited empirical evaluation on 1 language modeling task: also evaluating on tasks other than language modeling (say downstream tasks like the SuperGLUE benchmark, or Machine Translation) would significantly strengthen the claims made in the paper.

Questions or comments:
1. The idea of disentangling input dimensions from order to reduce kernel computation costs sounds very similar to depthwise separable convolutions [1].
2. Is the x-axis in Figure 5 mislabeled? From the caption and the text I would expect it to represent number of training steps or number of tokens trained on.

References:
[1] Xception: Deep Learning With Depthwise Separable Convolutions, Chollet et al.

**Summary Of The Paper:**

This work introduces Legendre Memory Units, an approach that utilizes non-parametric linear time-invariant layers to compute representations of a sequence of inputs. The key idea is similar to convolutions, but the weights of the convolution layer were static (non-trainable) and produced a representation of the sequence that can also be represented as a RNN. The LMU is equipped with a implicit self-attention layer; which bypasses the quadratic-time complexity of self-attention by performing attention over a fixed length sequence of hidden states.

The authors compare the proposed LMU against transformers on a language modeling task with the Webtext-2 dataset. This comparison is conducted over a range of model sizes (ranging from 55K to 1M non-embedding parameters), demonstrating that LMU improves over transformers with a similar number of parameters over the entire range; and extrapolating from the scaling law suggests that the difference would persist at larger model sizes.

By equipping the LMU with a global self-attention layer (or a transformer self-attention layer), the authors demonstrate additional gains similar to what was observed with a LMU over a vanilla transformer.

**Summary Of The Review:**

This work proposes a very interesting application of a novel, parameter-efficient architecture on a large-scale language modeling task. However, the paper presents an incomplete comparison that plays to the strength of the LMU architecture by comparing only parameter-matched models. Given that a significant fraction of the LMU compute is hidden in non-trainable parameters, a fair comparison should also compare models trained with an equivalent amount of training computation; ideally in terms of total training compute used.

Another weakness pertains to the limited number of tasks used for this comparison; also evaluating on tasks other than language modeling (say downstream tasks like the SuperGLUE benchmark, or Machine Translation) would significantly strengthen the claims made in the paper.

Overall, given the above two limitations I would recommend rejecting the paper. While the proposed architecture is interesting and well motivated, the paper needs stronger empirical validation and a fairer comparison against existing architectures.

---

> ### Author Response · Authors · 2021-11-22
> **Response 2**
>
> We thank the reviewer for their comments.
>
> >> The paper presents an incomplete comparison that plays to the strength of the LMU architecture by comparing only parameter-matched models. Given that a significant fraction of the LMU compute is hidden in non-trainable parameters, a fair comparison should also compare models trained with an equivalent amount of training computation; ideally in terms of total training compute used.
>
> We would like to point out two things here:
> 1) LMU models do not make use of positional embedding parameters, and for the transformer models, the parameter count excludes the positional embedding parameters (given by embed dimension * sequence length). For our largest model, the total number of non-parametric weights inside all the LMU layers (given by order * order + order) amounts to about 70% of the positional embedding parameters.
>
> 2) The total amount of compute per time-step that LMU models use is approximately the same as that of transformers. We attempted to match compute across all experiments.  For example, the largest LMU and transformer models both use ~24 * 10^6 operations at each time-step.
>
> We note this in the updated version of the paper.
>
> >> Limited empirical evaluation on 1 language modeling task: also evaluating on tasks other than language modeling (say downstream tasks like the SuperGLUE benchmark, or Machine Translation) would significantly strengthen the claims made in the paper.
>
> We agree, and we wish to evaluate our model on downstream tasks in the future. However, we would like to point to a previous work (Chilkuri & Eliasmith (2021)) that effectively used an LMU-based model (without any attention modules) to achieve better performance than DistilBert on the task of sentiment analysis.  Thus, our focus here is on more general scaling features than specific downstream tasks.
>
> >> Is the x-axis in Figure 5 mislabeled? From the caption and the text I would expect it to represent number of training steps or number of tokens trained on.
>
> The x-axis label (‘Parameters’) in Figure 5 is correct. It is similar to Figure 3, except that all the transformer models in this figure use 10x more training compared to the LMU models.

---

### Official Review · Reviewer_hsfs · 2021-11-03

**Correctness:** 3
**Technical Novelty And Significance:** 2
**Empirical Novelty And Significance:** 1
**Recommendation:** 3
**Confidence:** 4

**Main Review:**

Strengths:
- The output of the LMU can compress past history at each time step. The use of LMU reduces the self attention complexity from n^2 to q'xq where q is the order of the q Legendre polynomials. This provides advantage to tasks of larger sequence lengths.

Weaknesses:
- Like discussed in the paper, LMU's implicit self-attention is good at prediction with limited context, while the traditional self-attention can capture long-range dependencies. Also, LMU lacks a mechanism to provide pairwise information, which can be captured by self-attention. This can be illustrated using entailment tasks or translation tasks.

- Empirical results are very limited. Cross-entropy scores are limited because language modeling or a masked language modeling does not reflect how good the model can be at tasks that requires pairwise information. Better to compare with Transformer on down stream tasks with finetuning as well.

- Lacks a comparison with stronger baselines of efficient transformers. Better to compare agains Linformer (low-rank), Synthezer (Random generation), and Primer (learning the primitives).

- Lacks ablation study on the selection of window size and order in the LMU. q is only effective when sequence length is larger than q. For  tasks of smaller sequence length, they won't benefit from the LMU method.

[1] Linformer: Self-Attention with Linear Complexity

[2] Synthesizer: Rethinking Self-Attention in Transformer Models

[3] Primer: Searching for Efficient Transformers for Language Modeling


**Summary Of The Paper:**

The paper proposed a way to address two limitations of a transformer network. The method is based off a non-parametric Liner Time-Invariant component of the Legendre Memory Unit (LMU), which projects a sliding window of the input sequence onto a Legendre polynomials to provide a temporal representation and compression of the input signal. The newly proposed attention operates only on the output of the LMU at each time step, not across all time steps. The model is validated by studying the scaling properties of the models. The model is demonstrated scaling better than transformers or requires 10x fewer tokens to match the loss of a transformer network.

**Summary Of The Review:**

To sum up, the paper is too immature to publish in ICLR for several reason:
- First, it lacks a rigorous analysis on complexity compared to vanilla transformer, on various sequence length. Varying the sequence length, the performance of a LMU based method can be changed.

- The paper is weak in baseline. There are many recently work towards building efficient transformers, using low-rank, approximation, kernel based method, and neural architecture search. All of the work provide stronger performance than vanilla transformer. It is better to compare with at least a few of them.

- Perplexity or cross-entropy on a pretraining task such as LM or MLM can be very limited. Plenty of related work (such as gMLP [4]) have been demonstrated useful in the pretraining task, but they fail to outperform transformer in the down stream tasks with finetuning. Many work failed to provide pairwise information as a self-attention can provide.

[4] Pay Attention to MLPs

---

> ### Author Response · Authors · 2021-11-22
> **Response 1**
>
> We thank the reviewer for their comments.
>
> >> First, it lacks a rigorous analysis on complexity compared to vanilla transformer, on various sequence length. Varying the sequence length, the performance of a LMU based method can be changed.
>
> The performance of our model does depend on the length of the sequence. As we have listed in Table 1, the parallel implementation’s memory and compute scale as O(n) and O(n ln n) respectively, and the recurrent implementation scales as O(1) and O(n). Additionally, in Table 2, we list the total number of FLOPs, which is also a function of the sequence length (n), the order (q) and the reduced order (q’), and the total FLOPs for one block is given by:  3d (5(q’ + 1)(log(n) + 1) + q’) + 6qq’ + 2dq’^2 + 2dq’^2 + dq’ + 2dq’ + 8dd’.  We believe this constitutes a rigorous complexity analysis.
>
> >> q is only effective when sequence length is larger than q. For tasks of smaller sequence length, they won't benefit from the LMU method.
>
> Size of q is upper bounded by the length of the sliding window, theta, and theta in turn depends strongly on the total length of the sequence -- we have been using theta = sequence length / number of layers in the model. Thus, q is always less than or equal to the length of the sequence, with equality holding in the case where the model contains only one LMU layer.

---

> > ### Comment · Reviewer_hsfs · 2021-11-28
> > **Not changing my score**
> >
> > Thanks for authors for providing additional analysis on complexity and scaling. However, as discussed in the review, the are many unaddressed problems in the paper. The reviewer would encourage the authors continuing the effort and addressing those questions for a future acceptance.

---

### Author Response · Authors · 2021-11-22
**Overall Response**

We thank all the reviewers for their comments. There were a number of reviews that identified similar elements of concern. As a result, we address those as a group here, and then discuss additional individual concerns of the reviewers after that.

1. Comparisons with other efficient transformers (e.g. “Lacks a comparison with stronger baselines of efficient transformers. Better to compare against Linformer (low-rank), Synthezer (Random generation), and Primer (learning the primitives).” and “From the evaluation, it seems that comparing to other efficient transformers was not the main focus of the work but people might wonder how well it works compared to alternatives. What recommendation the authors would give for those interested in using it?“)

The efficient transformers, thanks to their lower memory requirements, can be trained on batches of extremely long sequences and this proves advantageous for tasks that benefit from long context, such as summarization and question answering. While the efficient transformers tend to outperform the original transformer in such settings, the original variant maintains better performance when working with typical sequence lengths. Since we are dealing with seq_len=1024, we reasoned that it would be most appropriate to compare against models employing full self-attention and not an approximate version. In short, we are demonstrating advantages where efficient transformers do not.

We note this in the updated version of the paper.

2. Ablation studies (e.g., “What is the contribution of each specific design choice, such as the FFN/global attention and implicit self-attention, to the end performance? An ablation analysis would be most appropriate for quantifying this.” and “In general, some ablations would make the paper much stronger and shed more light on the proposed model (on top of what is mentioned above, I would also like to see the effect of the values of q and q').”

We strongly agree with the importance of ablation studies for determining in more detail what aspects of the current architecture are most important for the measured performance. Unfortunately these could not be undertaken in the time between submission and response, given the significant time and expense of running the required experiments. We note that the simulations reported in the paper cost several tens of thousands of dollars to complete. This cost, along with the focus of the paper on providing a first demonstration of the important role LTI systems can play in NLP, lead us to view ablation as a more future-oriented line of work.

---

> ### Public Comment · ~James_A_Bowery1 · 2023-03-02
> **Consider The Hutter Prize.  It Limits Resources of Contestants.**
>
> "Unfortunately these could not be undertaken in the time between submission and response,
> given the significant $\textbf{time and expense}$ of running the required experiments. We note that the simulations reported in the paper cost several tens of thousands of dollars to complete."
>
> Please consider submitting an entry for The Hutter* Prize for Lossless Compression of Human Knowledge.  It is designed for resource-limited contestants so as to encourage $\textit{radical}$ innovation in machine learning under the most principled** information criterion for model selection: approximating the Algorithmic Information of a corpus of data (ref Solomonoff).  In this case the corpus is a 1GB snapshot of Wikipedia.
>
> Contestants are restricted to using up to: 50 hours on 1 CPU, 10GB of RAM and 100GB HDD.
>
> The current benchmark is about 115MB including the de/compression program and parameters required to generate the corpus without loss.
>
> *Marcus Hutter is the PhD advisor for DeepMind founders and is sponsoring this out of his own pocket.
>
> **The FAQ explains why approximation of Algorithmic Information (lossless compression) is the most principled model selection metric.  The fact that it makes virtue of a necessity fits perfectly with the parsimony claimed for the LMU.

---

### Decision · Program_Chairs · 2022-01-20

**Decision:**

Reject

**Comment:**

The paper proposes a language modeling architecture based on the RNN cells leveraging Legendre memory units. The proposal is interesting, but as all the reviewers notice, the paper is not ready for the presentation in the top ML conference for several reasons: comparison with weak baselines, shallow or weak analysis of the presented results, insufficient discussion of the related work, etc. Looking forward for all the comments to be addressed by the authors.

In the rebuttal the authors addressed some of the questions but all the reviewers think that the paper is not ready for acceptance and careful rewriting is needed. Recent research on the improved RNN mechanisms suggests that Legendre memory units and related mechanisms might be a gateway to solving several standard issues of training regular RNNs so the topic is definitely of great importance. Thus the authors are highly encouraged to resubmit the paper after making all suggested corrections.